# Laser Capture Microdissection: A Gear for Pancreatic Cancer Research

**DOI:** 10.3390/ijms232314566

**Published:** 2022-11-23

**Authors:** Bhavana Hemantha Rao, Pavel Souček, Viktor Hlaváč

**Affiliations:** 1Biomedical Center, Faculty of Medicine in Pilsen, Charles University, 306 05 Pilsen, Czech Republic; 2Toxicogenomics Unit, National Institute of Public Health, 100 00 Prague, Czech Republic

**Keywords:** Laser Capture Microdissection (LCM), pancreatic cancer, intraductal papillary mucinous neoplasm (IPMN), single-cell separation, hypoxia, metastasis

## Abstract

The advancement in molecular techniques has been attributed to the quality and significance of cancer research. Pancreatic cancer (PC) is one of the rare cancers with aggressive behavior and a high mortality rate. The asymptomatic nature of the disease until its advanced stage has resulted in late diagnosis as well as poor prognosis. The heterogeneous character of PC has complicated cancer development and progression studies. The analysis of bulk tissues of the disease was insufficient to understand the disease, hence, the introduction of the single-cell separating technique aided researchers to decipher more about the specific cell population of tumors. This review gives an overview of the Laser Capture Microdissection (LCM) technique, one of the single-cell separation methods used in PC research.

## 1. Introduction

Globally among the rare lethal cancers, pancreatic cancer (PC) ranks seventh in cancer-related deaths [1,2]. Recent advancement in imaging methods and treatment strategy has slightly improved the 5-year survival rate after diagnosis from less than 5% to less than 10% [3]. The pancreas is a mixed organ that has endocrine and exocrine functions, hence the tumors of the pancreas are also categorized accordingly [4]. Exocrine tumors, which can also be benign or malignant, make up 98% of them. Adenoma, cystadenoma, lipoma, fibroma, hemangioma, lymphangioma, and neuroma are a few benign tumors, whilst ductal adenocarcinoma is the most prevalent malignant tumor. Pancreatic ductal adenocarcinoma (PDAC) arises from non-invasive precursor lesions such as pancreatic intraepithelial neoplasm (PanIN), intraductal papillary mucinous neoplasm (IPMN), and mucinous cystic neoplasm (MCN) [4,5]. Based on the epithelial abnormality, the precancerous lesions are further categorized as depicted in Figure 1.

On the other hand, the remaining 1–2% of PC is caused by pancreatic endocrine tumors, otherwise known as pancreatic neuroendocrine neoplasms (PNENs), which are further categorized as pancreatic neuroendocrine tumors (PNETs) and neuroendocrine carcinomas (PNEC) [6]. PNENs mostly emerge from abnormal endocrine cells or the pluripotent cells of the pancreas. Insulinoma, gastrinoma, glucagonoma, vasoactive intestinal peptide (VIP)-oma, and somatostatinoma are some of the endocrine tumors recognized today. They are classified as functional or non-functional, based on the hormones released by the tumor [7]. The heterogeneity of the disease is one of the reasons for the complications faced in PC research. Like any other cancer, PC progression is also supported by its microenvironment. The tumor microenvironment consists of both cellular and non-cellular components such as stromal cells, fibroblasts, immune cells, and signaling molecules, along with the tumor cells that aid in cancer advancement [8]. Pancreatic stellate cells, the most abundant in stromal cells that provide nourishment; the extracellular matrix (ECM) proteins such as laminin, fibronectin, proteoglycans, glycoproteins, and polysaccharides; and cancer-associated fibroblasts (CAFs), an important factor involved in stromal-to-tumor interaction are the major contributors to the PC development [8]. Apart from the different cells present in the tumor microenvironment, PC also consists of different cancer subtypes. Based on the transcriptomic studies, the PC subtypes are classified as normal stroma and activated stroma; basal-like and classical-like tumor cells, by Moffitt et al. [9]. Collisson et al. classified them as classical, quasi-mesenchymal, and exocrine-like subtypes [10]. Bailey et al. categorized them as squamous, progenitor, immunogenic, and aberrantly differentiated endocrine exocrine (ADEX) [11]; whereas Puleo et al. redefined Moffitts’s classification into pure basal-like, stroma-activated, desmoplastic, pure classical, and immune classical [12]. Comparing these classifications, basal-like, quasi-mesenchymal, squamous, and stroma-activated, have the same lineage of origin, that is squamous; the classical, exocrine-like, and ADEX was similar in exocrine functions, whereas the immune classical and immunogenic were found to have roles in immunologic functions [13].

Cancer development is always due to the accumulation of multiple mutations in cells along with other triggers, such as environmental and lifestyle factors. Various genetic alterations such as the mutation activation of oncogenes (example: KRAS), the inactivation of tumor-suppressor genes (example: CDKN2A, TP53, SMAD4, BRCA2), the loss of heterozygosity during gene amplification, and telomere shortening, contributes to this [5,14]. Hence, understanding these changes would help in developing better tools for diagnosis, prognosis, and therapy. The limited diagnostic methods of imaging and biopsies, the deep inside location of the pancreas, and the asymptomatic nature until the advanced stage has challenged the diagnosis, as well as the prognosis, of the disease [15,16]. Today, physicians depend on biomarkers such as CA 19-9, CA 50, and CEA for diagnosis, to determine the response to treatment, as well as for the prognosis of the recurrence of PC, and they have shown low sensitivity and specificity [17,18]. Research on biomarkers associated with PC is being extensively carried out, but none has implemented in clinical practice due to a lack of validation [18]. It is possible for the proto-oncogenes, tumor suppressor genes, and other genes (DNA) expressed during cancer conditions, as well as coding (mRNA) and non-coding RNAs (miRNA, siRNA, lncRNA, etc.), and proteins to act as diagnostic, prognostic, or predictive biomarkers [18]. They can be isolated from cancer tissues, circulating tumor cells (CTCs), and exosomes present in body fluids such as saliva, pancreatic juice, blood, urine, and stool [17,19]. 

Nowadays, cancer researchers are interested in high throughput techniques, especially emerging single-cell sequencing technology (SCST), which involves sorting the cells before sequencing. Most of the complications in the diagnosis, prognosis, and therapy of PC are resolved by this technology [20]. SCST involves four major steps: the isolation of individual cells; the amplification of the nucleic acid; the interrogation of the amplified products; and the interpretation of the data, considering all the biases and errors that occurred in the first three stages that are more critical in producing high-quality data [21]. A cell suspension of viable single cells is prepared by mechanical or enzymatic methods. Earlier, patch-clamp electrophysiology, fluorescence in situ hybridization, flow cytometry, and ELISpot were some of the methods available to isolate and examine the single cells [22] but now, methods like Laser Capture Microdissection (LCM), Fluorescent Activated Cell Sorting (FACS), and Microfluidics technologies are commonly used for the single-cell isolation, mainly from cell suspensions, except for LCM [23]. The nucleic acids and proteins isolated from the sorted cells or a specific cell population are then lysed using optical, ultrasonication, electrical, mechanical, or chemical methods [24]. The isolated biomolecule of interest from the cells are then subjected to amplification and analysis for which methods like DOP-PCR, isothermal amplification (multiple displacement amplification (MDA), Microwell displacement amplification system (MIDAS)), and hybrid methods (MALBAC or PicoPLEX) are employed for whole-genome amplification. The amplified products can be explored by either sequencing specific loci, sequencing the whole-exome, or the entire genome [21]. For single-cell transcriptomic studies, the methods such as reverse transcription, quantitative real-time PCR (qPCR), microfluidics platforms like CytoSeq, inDrop, and DropSeq techniques, are used [22,24]; whereas 2D-gel electrophoresis, mass spectrometry, Edman sequencing, and NMR spectroscopy techniques are utilized for proteomic studies [25]. The CytoSeq microfluidic method is a recent technology that includes magnetic beads containing random primers with a universal PCR priming site, a barcode, a unique molecular index (UMI), and an mRNA capture sequence that helps in yielding high-quality, bias-free, and quantitative data of single-cell transcriptome, whereas other two microfluidics methods are automated in which the nanodrop of cells formed from the oil and water are barcoded, lysed and sequenced and referred as droplet-based microfluidics; this method is also known as a lab-on-a-chip method [22,26]. In droplet-based microfluidics, the cells in the suspension are generated into droplets with the help of passive droplet fusion, electro-coalescence, or picoinjection, and then sorted [27,28]. The principle of droplet-based microfluidics is employed in FACS for the fluorescent-labelled samples and further sorted using the hydrodynamics-optic principle [29]. The examination of the amplified product is one of the crucial steps in single-cell analysis; the data from the analysis are interpreted using a combination of different biostatistical and bioinformatics tools [30]. 

Having high-throughput sequencing technologies, LCM marks its importance in understanding the role of individual cells/a specific cell population in a heterogeneous population of cancer tissue, thus providing precise information compared to the results from bulk tissues [31]. Likewise, LCM is being used in numerous research projects to explore PC. Hence, this review gives an insight into the application of laser-assisted microdissection techniques in various aspects of PC research.

## 2. Laser Capture Microdissection (LCM) 

LCM is a sophisticated technique in which a laser is coupled with an inverted microscope and linked to a computer. This technique was developed in 1996 at the National Institute of Cancer, USA, for isolating selected human cell populations from a heterogeneous population of cells (Figure 2) [32]. The laser systems used in this apparatus have been modified since then from ultraviolet (UV) to high-energy nitrogen, infrared, and carbon dioxide lasers [33]. Based on the laser beams used, LCM can be of two types: infrared and UV. The commercially available Arcturus Pixcell IIE LCM platform is an example of infrared LCM, whereas the PALM Laser-MicroBeam System, MMI cellcut^®^ (Molecular Machines and Industries), and the Leica Laser Microdissection system, are commercialized instruments in which UV-laser beams are installed [31,33,34,35]. The latter is a widely used instrument in which the solid tissues are typically prepared on a membrane-covered slide to identify the target cells and examined under the microscope [32], whereas the cell culture specimens are cultured on membrane-bound culture plates to which the cells adhere [36]. The membranes are usually polyethylene naphthalate, to which methods such as hematoxylin/eosin staining, fluorescent in situ hybridization (FISH), and immunohistochemistry (IHC) are used for staining the cells [33]. Then, the cells of interest are manually marked on the computer screen, and the laser cuts along the marked direction [32]. This is followed by contact-based extraction, contact-free gravity-assisted microdissection (GAM), or contact-free laser pressure catapulting (LPC), after which the cells are treated with appropriate buffers for processing and sequencing [26]. While working with the frozen tissue sample, careful handling during the cryosectioning, staining, and marking of the cells of interest for microdissection aids in yielding a high-quality nucleic acid for next-generation sequencing [37]. 

Like every technique, LCM has benefits and drawbacks. It enables small tissue isolation from a heterogeneous population in a single step through direct visualization through a microscope; it is a fairly quick method of dissection; it secures the tissue morphology during the dissection; and it also allows to separate live cells/single cells in a culture dish and re-culture them. Especially when the content of tumor cells in neoplastic tissue is low, LCM takes the advantage of enriching the tumor fraction. The drawbacks of this method include its high cost, the need for a histologist or other specialized personnel to identify the cells of interest, and the possibility that the quality of the dissected tissue will not meet the standards needed for the further processing of the sample because of the absence of a coverslip, which causes dehydration of the sample [33,35]. When compared to FACS and microfluidics systems, which are often used on liquid samples especially for separating cells from the blood, LCM can isolate the single cells from tissue samples like FFPE and fresh frozen, whereas the need for professional personnel to identify the cells holds the disadvantage of LCM. In addition, though it is time-consuming, the tumor cells can be isolated using a laser beam directly without treating the samples with fluorophores, which is often done in both FACS and microfluidics [38,39]. Having advantages and disadvantages of the technique, LCM is an efficient tool for isolating single cells from tissue samples.

The application of LCM in cancer research was reviewed in some solid cancers. Lawrie and Curran [40] described the use of LCM in colorectal cancer proteomics; Fuller et al. [31] reviewed the utility of LCM in breast cancer; neuroblastoma [41] and prostate cancer [42] were also reviewed. However, this was before the onset of novel techniques such as next-generation sequencing. Some recent reviews advocate the use of LCM in oral cancer [43] or testicular germ cell tumors [44]. Liotta et al. [45] described the use of LCM in the protein analysis of solid cancers and methodology, with example applications in cancer tissues, thoroughly reviewed in von Eggeling and Hoffmann [46]. However, the whole omics view on the use of LCM in PC is completely missing.

**Figure 2 ijms-23-14566-f002:**
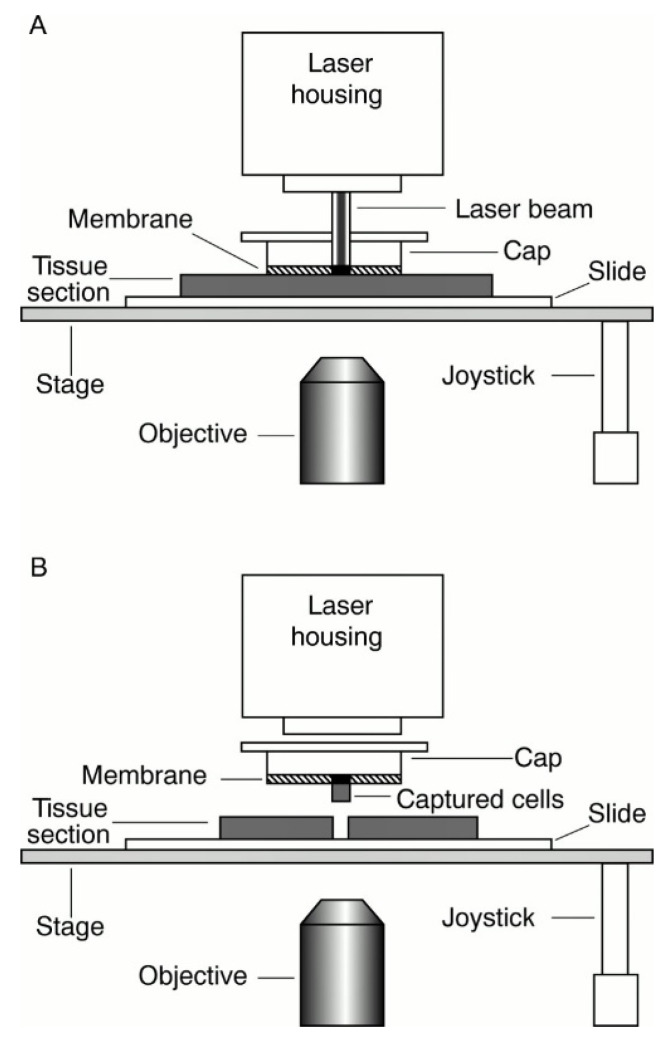
Schematic diagram of LCM: (**A**) in-house laser capturing the samples in marked trajectory using the laser beam; (**B**) the cap lifting the captured samples in the slides [47]. Reproduced from laser capture microdissection in pathology, Falko Fend and Mark Raffeld, Vol:53:666–672; 2000 with permission from BMJ Publishing Group Ltd.

## 3. Impact of LCM on Pancreatic Cancer Research

The heterogeneous cell population and numerous genetic alterations resulting in gene expression differences contribute to PC’s aggressive nature. Exploring these areas aids in developing new biomarkers for improvement of the treatment and maintaining the condition [5]. The research work has been carried out extensively in elucidating the mutations, understanding the proteome, as well as the prognostic and predictive characterization and stratification of the PC patients, which is discussed in the following sections.

### 3.1. Mutation Studies 

The heterogeneous nature of PDAC is characterized by various genetic alterations like the activation of the proto-oncogene, *KRAS*; the inactivation of *CDKN2A*, *TP53*, *SMAD4*, and *STK11*/LKB1, the tumor suppressor genes [48]; the loss of heterozygosity at 19p13.3 [14], 6q and 17p (in IMPNs) [49], and so on. These alterations are thoroughly studied to understand their role in PC metastasis, seek their role in prognosis/survival, or identify therapeutic targets [14]. Before the use of single-cell separation methods, the bulk tissues were analyzed for the research in PC. The mutations of *KRAS* and *TP53*, inactivation of p16/*CDKN2A*, and *SMAD4*/DPC4 in PanIN, IPMN, and MCN were found in the analysis of FFPE samples of tumor tissues [50].

*KRAS* mutation was found to be an early event in all three precancerous lesions accompanied by p16/*CDKN2A* inactivation in PanIN and IPMN. Whereas *TP53* mutation and silencing of *SMAD*/DPC4 were the late events in PanIN, IPMN, and MCN [50]. Also, the frequent mutation of *KRAS* at codon 12 and exceptionally at codon 13, and 61; the difference in the pattern of *KRAS* mutation in Japanese and European populations with GGT to GAT (G12D) in Japanese and GGT to GAT (G12D), GGT to GTT (G12V), CGT (G12R) or TGT (G12C) in European population respectively were discovered before the introduction of LCM [50].

Using LCM, most of these findings were confirmed like the typical *KRAS*, *SMAD*/*DPC4*, and *TP53* mutations, along with the somatic mutation of *PIK3CA* in MCN [51], and along with it, many more interesting facts were deduced from the specific cell population of the tumors such as the mutation analysis conducted by Crnogorac-jurcevic et al. two decades ago using LCM treated normal and tumor samples (the overview of the LCM-based studies is provided in Table 1) of PDAC revealing the homozygous deletion of *CDKN2* and the mutation of *KRAS* using single-strand conformation polymorphism (SSCP) and direct sequencing methods. In the same experiment, they used a cDNA array, tissue array, and IHC to discover the involvement of overexpressed genes *TIMP1*, *CD59*, *ABL2*, *NOTCH4*, *SOD1*, and the downregulation of *XRCC1* gene in different pathways leading to pancreatic malignancy [52].

Similarly, the population-based study conducted on the Japanese and European populations was performed on Chinese populations using LCM, PCR, and direct sequencing. The *KRAS* mutations in the Chinese population were found to be different compared to the Japanese-European population with a mutation in the first or second base of the codon 12 (GGT) [53]. A similar study on *KRAS* and *TP53* gene mutation in PDAC patients from highly polluted regions of the Nile River delta in Egypt to the less polluted region showed a significantly higher rate of mutation in the *KRAS* codon 12 G to T (G12V) transversion mutation and mutation in exon 5-8 of TP53 in patients from highly polluted areas, pointed out the importance of the interaction of environment and genes in carcinogenesis [54].

A different study was carried out by Izawa et al. [55] on LCM-derived IPMN tissues to study the clonal characterization using the combination of *KRAS* analysis and analysis of human androgen receptor gene (HUMARA) during X-chromosome inactivation. The study concluded the polyclonal/oligoclonal nature of IPMNs and their origin from multiple precancerous lesions. Pancreas is made up of different cells like ductal, stellate, acinar, and beta cells among which, acinar cells perform the role of secreting digestive enzymes namely amylase, protease, and lipase in the form of zymogens [56].

The analysis of the whole-tumor tissue had given researchers the idea that acinar cells could be the origin of human pancreatic neoplasia but with the help of LCM, PanIN lesions, acinar-ductal metaplasia lesions, stromal cells, and acinar cells were isolated and closely studied for *KRAS* mutation (LigAmp technique) to disapprove this hypothesis [57]. There were contradictory findings like the study using LCM showing the *TP53* gene could be found in the early stage of PDAC, which was found to be a late-stage event by the study on bulk tissue, but the same experiment supported that *KRAS* mutation along with other somatic gene mutations found in early-stage PanIN-2 lesions promotes the PDAC progression [58]. However, a recent study substantiated the role of *TP53* in the evolution of PDAC and found that *TP53* is not only a gateway to genetic chaos but also a provider of deterministic patterns of genome evolution that may show new strategies for the treatment of tumors with *TP53* mutation [59].

In addition, the study conducted by Fang et al. [60] on pancreatic adenosquamous carcinoma (PASC) and PDAC samples dissected using LCM revealed the possibility of the origin of both cancers from the same progenitor cancer cells, that was supported by the similar results in the genomic variation in *KRAS* and *TP53* genes. They also highlighted the importance of 3p loss in PASC with the help of copy number variation analysis that gave the researchers new insight into the knowledge about mixed-type tumors. Another type of mixed tumor, acinar-neuroendocrine-ductal carcinoma, was rarely diagnosed and studied with the help of LCM for isolating the acinar and neuroendocrine tumor cells for performing next-generation sequencing. This type of cancer was found to have a frameshift mutation in *TP53* (p.N210fs) and a missense mutation in *KRAS* (p.G12R) [61].

Carcinosarcoma is also one among the rare PC, and the study performed by Bai et al. [62] with the help of LCM to isolate carcinomatous and sarcomatous cells from carcinosarcoma samples, were studied for IHC, clinicopathological, and *KRAS* mutation, which showed similar mutation pattern in *KRAS* mutation (p.G12D and p.G12V) in both the samples, indicating that both the components has a monoclonal origin.

The conflict in the results from bulk and specific cell population also includes the study of epidermal growth factor (*EGFR*) that plays a critical role in many cancer types, with its downstream pathways including RAS-MAPK/PI3K-AKT-mTOR pathways that are well studied in cancer prognosis [63]. However, the study on pancreatic cell lines and clinical samples using LCM and direct sequencing revealed that the *EGFR* gene is highly conserved in pancreatic cancer and contradicted its association with PC prognosis, leaving room for other explanations of the relevance of *EGFR* mutation in PDAC [64]. Additionally, one of the simulation studies performed by Fujii et al. [65] demonstrated the lack of microsatellite instability in PC using specialized fluorescent microsatellite analysis on microdissected PDAC specimens, but the study showed profound LOH in these samples. Further, they recommended against the usage of LCM in microsatellite instability studies using tissue samples.

Several other fascinating observations about *KRAS*, such as the study demonstrating the carcinogenic role of the secretory and trophic effects-regulating hormone, gastrin, was carried out on gastrin gene-knockout, *KRAS*-mutant mice, and in human samples microdissected using LCM. The results were interesting as the knockout mutant mice showed decreased PanIN progression, inflammation, and fibrosis compared to the results obtained from the re-expression of gastrin. The decrease in KRAS expression reverted the signal transduction to the canonical pathway and they found a significant increase in the gastrin mRNA expression in PC samples when it was re-expressed. Hence, with the help of LCM on healthy pancreatic tissue and tumor tissues, the expression study of gastrin unveiled its possible role in activating *KRAS* in PC [66].

Likewise, other findings such as the silencing of *CDKN1C* by an epigenetic mechanism [48], the role of *FXYD3* in cell proliferation [67], and the association of the Sox4/Ezh2 and miR-335 with the epigenetic mechanism of Sox4 expression, which in turn stimulated EMT pathway [68], were possible on cell lines, xenografts, and PDAC samples using LCM as it aided in the selection of tumor and healthy tissue. These studies employed techniques like DNA oligonucleotide microarray, IHC, qPCR, semi-quantitative reverse-transcription PCR, methylation-specific PCR, northern blot, immunofluorescence, and bisulfite sequencing, which is further detailed in Table 1. The study by Hasegawa et al. [68] also found an association of miR-335 with poorer disease-free and overall survival. Nakahara et al. [69] found the possibility of miR-101-*EZH2* interaction in microdissected IPMN samples to play a vital role in IPMN carcinogenesis. The knock-down study of miR-101 on PC cell lines confirmed that the miRNA targets *EZH2* and the loss of miR-101, which was effectively found in most of the IPMN samples, could stimulate the PDAC by upregulating *EZH2*. Hence, the study highlights the therapeutic target efficiency of miR-101.

### 3.2. Breakthrough of PC Subtypes and Their Relevance in Survival

The integration of genomics, transcriptomics, proteomics, methylation studies, and other omics studies, can help better understand and identify biomarkers of early diagnosis, prognosis, or therapy prediction of cancer patients. It also helps to identify the targets for treatment. The study on molecular subtypes of pancreatic cancer contributed to understanding the survival of patients. Collisson et al. used LCM-based techniques to distinguish cancer and stromal subtypes of PC [10]. Thereafter, Moffitt et al. [9] determined the PC subtypes using an algorithm-based virtual microdissection on PDAC tissue samples and validated the use of bulk RNA-sequencing data using the Non-negative matrix factorization (NMF) method. A similar study was conducted by Kalloger et al. [70]. This study suggested the prognostic roles of the genes *KRT6A*, *CTSV*, and *LY6D* and highlighted the urge for studies on stromal cells and their importance in cancer progression compared to the studies in epithelial PDAC. Puleo et al. [12] preferred manual microdissection and supported the work of Moffitt et al. [9] whereas Bailey et al. [11] and Maurer et al. [71] conducted the genomic analysis and RNA sequencing, respectively, on bulk tumor tissue, which resulted in more subtypes. In short, the clarity on molecular subtypes of PC was attained using the single-cell separation method.

Recently, Birnbaum et al. [34] took an effort to conduct a transcriptomic study to explore the role of PC subtypes of cancer and stromal cells in prognosis and precision medicine. They identified four cancer subtypes (C1–C4) and three stromal subtypes (S1–S3) and they correlated it with the short-term survival and long-term survival using differentially expressed gene (DEG) analysis. The canonical pathway and Gene Ontology (GO) biological process evaluated the involvement of the C1 subtype in protein folding and leukocyte chemotaxis; C2 in neuronal membrane signaling and pancreatic endocrine cell development; C3 in protein translation regulation and nucleotide biosynthesis; C4 in the oncogenic signal transduction pathway; S1 in cell development and differentiation; S2 in antigen processing and presentation; and S3 in macromolecular modification. These sub-types were identical to Bailey, Collison, Moffitt, and Puleo’s classifications, in which, C1 and C3 were found similar to classical or pancreatic progenitor subtypes, C2 to ADEX, or exocrine-like subtype, and C4 to squamous or basal-like or quasi-mesenchymal subtype [9,10,11,34].

From the gene expression study conducted for prognosis and survival, genes associated with short-term survival were associated with cell plasticity, axon guidance, cell proliferation, and signal transduction; whereas long-term survival was associated with cell cycle regulation and tRNA/mRNA processing. Out of 113 genes, 13 genes were found to be exclusively expressed in cancer cells and they were confirmed by the two-color RNA-ISH (RNA-In situ Hybridisation). Genes *AP5M1, TCP1,* and *PNP* associated with long-term- and *MIA*, *MUC16*, and *ADGRF1* associated with short-term survival were highlighted as gene signatures for survival [34]. The microdissection technique was used for the investigations on subtypes and their association with survival within the recent prospective trial study, COMPASS, initiated at the Princess Margaret Cancer Centre in Toronto. Researchers used metastatic tumor cells to study the predictive mutational and transcriptional characteristics of PDAC for better treatment selection [72]. Whole-genome sequencing and RNA-sequencing of the microdissected samples revealed that the subtypes from the III/IV stage of PDAC were similar to Moffitt et al. classical subtype tumors and their response to mFOLFIRINOX first-line chemotherapy was better compared to the basal-like tumors. They also highlighted the importance of *GATA6* expression in differentiating the classical and basal-like subtypes in PDAC [72].

Apart from the cancer subtypes studies, the study conducted by Nakamura et al. [73] on DEG of different zones of same PC samples isolated from mice implanted with Human L3.6pl PC cells analysed using LCM, affymetrix GeneChip hybridisation techniques concluded that it is important to understand expression profiles of zonal heterogeneity in the discovery of prognostic and therapeutic biomarkers, and LCM aids in the reproducibility of the analysis in such studies.

### 3.3. Proteins, Pathways, and Cancer Management

Understanding the protein profile of the cancer is always of key importance, as it helps to enlighten the pathways leading to cancer development and metastasis [74]. The proteomic studies could be of two types: expression proteomics and functional proteomics. The studies that focus on the upregulation and down-regulations of proteins are the expression proteomics, whereas the studies that focus on the molecular mechanism and the unraveling of the biological functions of novel proteins are called the functional proteomics study [75]. Like the proteomic studies conducted on bulk tissues, which is not the scope of this discussion, single-cell separation methods, especially LCM, are also employed to understand the expression and functional proteomics of PC. Here, we highlight some of the proteins elucidated using LCM that are involved in various pathways of cancer progression.

In PC, there are several proteins such as the S100 family, a small integrin-binding ligand N-linked glycoprotein (SIBLING) family, and secreted protein acidic and rich in cysteine (SPARC) family proteins associated with cancer progression [76,77]. Bone Sialoprotein (BSP) is a member of the SIBLING family of proteins and was studied using LCM along with qPCR, DNA microarray, immunoblotting, radio-immunoassays, IHC, cell-growth, invasion, scattering, and adhesion assays on chronic pancreatitis, PDAC, and PC cell lines, to mark its importance in cancer growth, and metastasis (Figure 3) [78].

SPARC-like protein 1 (SPARCL1), a SPARC family protein, aka Hevin, found in the extracellular matrix was studied in LCM-applied normal pancreatic tissue and PDAC samples, using qPCR, and other protein analyzing methods. SPARCL1 expression was elevated in PDAC samples compared to the normal tissue and PC cell lines, and its expression was found to be downregulated in the late stages of PC indicating the role of SPARCL1 as a tumor suppressor gene (Figure 3) [79].

Similarly, the S100 family proteins are another widely studied, calcium-binding protein family, which has the potential to contribute to the early detection and prognosis of PC [76]. S100A6 was investigated with the help of LCM by A.R. Shekouh et al. They performed LCM with 2D-gel electrophoresis and other techniques such as isoelectric focusing, silver staining, MALDI-TOF, and IHC on normal and malignant tissue samples and validated that this calcium-dependent protein is highly expressed in tumor cells compared to the normal tissues [80]. The same combination of techniques, along with fluorescence dye saturation labelling, was performed on PanIN and normal samples along with comparing the data with proteome reference to find the role of three actin filament proteins (actin, transgelin, and vimentin) in PC progression [81]. S100P, a member of the same family, along with another protein 14-3-3 sigma/SFN, was found to be a promising biomarker in a study on PDAC and its matched lymph node metastasis FFPE sample microdissected using LCM (Figure 3) [82]. Later, a study conducted by F Robin et al. tackled the molecular profile of stroma from fresh frozen PDAC, separated using LCM, and analyzed using genome-wide expression profiling, tissue microarray, IHC, and ELISA to conclude that SFN/14-3-3 sigma/stratifin can be a potential candidate for the prognostic biomarker of PDAC [83]. It was clear that stratifin (14-3-3 sigma) played a vital role in cell cycle regulation and apoptosis using the combination of LCM, qPCR, DNA arrays, IHC, and western blotting [84]. The interesting fact is that these proteins stimulate the downstream main pathways like KRAS, apoptosis, DNA damage control, regulation of G1/S phase transition, Hedgehog, and many more [77,84], which gives them the potential to be used as diagnostic or prognostic biomarkers, as well as possible therapeutic targets.

Chronic pancreatitis (CP) is one of the risk factors for PC, a study comparing the protein expression in LCM performed CP, PDAC, and normal cells adjacent to infiltrating PDAC samples, were studied and deciphered the significant expression of cartilage glycoprotein-39 (HC gp-39), pancreatitis-associated proteins (HIP/PAP), and lactoferrin in both the samples compared to the healthy tissue indicating the potential role of these proteins as a predictive biomarker [85].

The study conducted by Sawai et al. [86], on one of the DNA editing enzyme, activation induced cytidine deaminase (AID), in the microdissected PDAC and normal tissue showed a significant increase in AID expression in acinar ductal metaplasia, PanIN, and PDAC suggesting the involvement of the protein in inducing cancer. It was further validated by deep sequencing the samples obtained from transgenic AID mice.

However, by employing LCM, researchers have made the initial step toward identifying several more proteins associated with PC that has not yet been fully studied [87,88,89,90,91,92,93], like the downregulation of Cav-1 as a possible prognostic marker in PC (included in Table 1) [89]. They have tried to understand the tumor progression using LCM along with proteomic studies that included LC-MS/MS, tissue microarray, and IHC on fresh frozen PDAC and adjacent normal tissues [88]. There are also studies using LCM (referred in Table 1) showing the influence of CTCs [90], lncRNA H19 [91], HOTTIP [92], and FN1-ITGA-3 [93] on PC prognosis, which has to be studied in detail for further clarifications.

**Table 1 ijms-23-14566-t001:** Overview of developments in PC research using LCM.

Author/Year	Finding	Sample Used	Techniques Used along with LCM	Reference
Emmert-buck et al., 1996	Discovery of LCM technique			[32]
Crnogorac-Jurcevic et al., 2002	Association of *ABL2*, *NOTCH4*, *SOD1*, *XRCC1* with metastasis of PC	Fresh frozen tissue of PDAC and PC cell lines (ASPC1, Bxpc-3, CaPan1, CaPan2, HS766T, Mia PaCa-2, PANC-1, SU86.86)	Micro-array derived gene expression analysis, quantitative real-time PCR (qPCR), Tissue array, IHC	[52]
Shekouh et al., 2003	Identification of DEGs in PDAC	Fresh frozen samples of PDAC and normal tissues	Isoelectric focusing, SDS-PAGE, silver staining, MALDI-TOF, IHC	[80]
Guweidhi et al., 2004	role of 14-3-3sigma/stratifin in cell cycle regulation, and apoptosis	Fresh-frozen and PPFE samples of human PDAC and normal tissues	cDNA array, qPCR, southern blot, IHC, mutation analysis (sequencing), western blot, immunoprecipitation, FACS analysis	[84]
Kayed et al., 2005	Role of *FXYD3* in PC development	FFPE samples of PDAC and PC cell lines (ASPC-1, BxPc-3, CaPan-1, Colo-357, SU86.86, T3M4)	qPCR, DNA oligonucleotide microarray, IHC, northern blot, immunofluorescence	[67]
Wei et al., 2005	The difference in *KRAS* mutation in the Chinese population	PDAC Samples	PCR and direct sequencing	[53]
Erkan et al., 2005	Role of BNIP3 in chemoresistance resulting in poor prognosis and survival in PDAC	PDAC tissue samples and PC cell lines (ASPC-1, BxPc-3, CaPan-1, Colo-357, MiaPaCa-2, Panc-1, SU86.86, T3M4)	cDNA microarray, qPCR, IHC	[94]
Sato et al., 2005	Down-regulation of CDKN1C in PC by an epigenetic mechanism	Fresh frozen IPMNs and normal tissues PC cell lines (AsPC1,BxPC3, CaPan1, CaPan2, CFPAC1, Hs766T, MiaPaCa2, Panc1), and xenografts	Microarray, semiquantitative reverse-transcription PCR, IHC, Methylation-specific PCR, and Bisulfite sequencing	[48]
Sitek et al., 2005	Role of actin filament proteins in PanIN progression	Fresh frozen PanIN samples and PC cell lines (CFPAC, CAPAN, Hs766T, IMIMPC-2, SCPC-1, PATH-8988T)	2-D electrophoresis (2-DE), fluorescence dye saturation labeling, MALDI-TOF	[81]
Fukushima et al., 2005	Role of HC gp-39, lactoferrin, and HIP/PIP as potential predictive biomarker of PC	Fresh frozen tissues and serum samples	Oligonucleotide hybridization, IHC, qPCR, ELIS	[85]
Hwang et al., 2006	Upregulation of PGK1 in PDAC and its potential role in therapeutic strategies or as a diagnostic biomarker	PDAC and normal tissues, serum samples	2-DE, MALDI-TOF, ELISA, IHC, Western blot	[95]
Tzeng et al., 2007	Conservation of *EGFR* in PC and its unavailability to act in the prognosis of PC	PDAC tissue samples, PC cell lines (S2-VP10 AND S2-103)	PCR, and sequencing	[64]
Kayed et al., 2007	Role of BSP in cancer progression	PDAC and chronic pancreatitis (CP) tissue, PC cell lines (ASPC-1, BxPc-3, CAnPan-1, Colo-357, MiaPaCa-2, Panc-1, SU86.86, T3M4)	qPCR, cDNA array, IHC, Radioimmunoassay (RIA), FACS, Invitro invasion, scattering, and adhesion assays	[78]
Esposito et al., 2007	Role of SPARC1 as a tumor suppressor gene in PC	Fresh frozen PDAC, PanIN tissue samples, and PC Cell lines (ASPC-1, BxPc-3, Capan1, colo-357, Su86.86, and T3M4)	FACS, in-vitro invasion assays, IHC	[79]
Soliman et al., 2007	Importance of gene-environment interaction in cancerogenesis	FFPE samples of PDAC and normal tissues	PCR, DNA sequencing	[54]
Nakamura et al., 2007	Importance of DEG studies in zonal heterogeneity of PDAC	Human PC cell line (L3.6pl), nude mice	Affymetrix HG-U133 plus 2.0 array, FISH	[73]
Hoffmann et al., 2008	Overexpression of *HIF1A* during the hypoxic condition in PDAC and its correlation with *PDGFA*, *VEGF*, and *FGF2*	PDAC FFPE samples	qPCR	[96]
Shi et al., 2009	Involvement of acinar cells in the development of PanIN/PDAC	PanIN lesions	PCR, LigAmp analysis, IHC	[57]
Kubo et al., 2009	Mutation of *KRAS*/*BRAF* in resequenced tyrosine kinase gene showing its importance in the downstream signaling pathway	PDAC samples and cell lines	WGA and sequencing	[87]
Collisson et al., 2011	Subtypes of PDAC	PDAC FFPE samples and Cell lines (HPAC, Capan2, HPAF II, 6.03, CFPac1, MPanc96, 2.13, Panc1, MiaPaca2, 10.05, and Colo357)	IHC, microarray	[10]
Kayashima et al., 2011	Stimulation of *INSIG2* in PC during hypoxia condition	PC cell lines (SUIT-2, ASPC-1, BxPC-3, PANC-1, KP-1N, KP2, KP-3, MiaPaCa2, CaPan1, CaPan 2, CFPAC-1, SW1990, HS766T, H48N, NOR-P1, HDPE6-E6E7) and PanIN lesions	qPCR, microarray	[97]
Naidoo et al., 2012	Protein composition of PDAC and lymph node metastasis	PDAC FFPE samples	Multidimensional Protein Identification Technology (MudPIT), IHC	[82]
Nakahara et al., 2012	Role of miR-101 as a therapeutic target in IMPNs	FFPE samples, PC cell lines (PANC-1, PK8, PK9, PK-59, KLM-1, MIA PaCa2, PK-45P)	IHC, qPCR, knock-down of miR101	[69]
Zhu et al., 2013	A better understanding of tumor progression using proteomic analysis of PDAC samples	Fresh frozen PDAC and adjacent normal tissue	LC-MS/MS, Tissue microarray, IHC	[88]
Murphy et al., 2013	Mutation of *KRAS*, *TP53*, and other somatic genes in PanIN-2 lesions and its role in PDAC progression	Frozen PDAC samples	Exome sequencing	[58]
Shan et al., 2014	Downregulation of Cav-1 as a prognostic indicator in PC	Fresh frozen PDAC samples	IHC, reverse-transcriptase PCR, qPCR, FISH	[89]
Garcia-Carracedo et al., 2014	PIK3CA mutation in pancreatic MCN	FFPE samples of MCN	IHC, direct sequencing	[51]
Sawai et al., 2015	Role of AID in PDAC development	PPFE samples of PDAC tissues, transgenic mice	IHC, deep sequencing	[86]
Hasegawa et al., 2015	Role of Sox4/Ezh2 in epigenetic mechanism and EMT pathway in PC patients	Fresh frozen PDAC samples	IHC, qPCR	[68]
Court et al., 2016	Role of CTCs in molecular diagnostics of PC	PC cell lines (CFPAC-1, ASPC-1, Panc-1, BxPC-3, HPAF-II) and blood samples of pancreatobiliary cancer patients	WGA, KRAS PCR, Sanger sequencing	[90]
M.Ling et al., 2016	Role of lncRNA H19 in PC tumorigenesis	Fresh frozen PDAC and normal tissues, PC cell lines (Colo-357, Capan1, MiaPaca-2, AsPC-1, BxPC-3, Panc-1, T3M4, SW1990)	qPCR, western blot, IHC	[91]
Fu et al., 2017	Role of lncRNA HOTTIP in DFS of PC	Cell lines (PANC-1 and SW1990)	qPCR, western blot, FACS, IHC	[92]
Fang et al., 2017	Showed PASC and PDAC originated from same progenitor cancer cells	FFPE samples of normal and tumor tissue	Whole-genome, whole-exome sequencing	[60]
Anug et al., 2018	COMPASS trial	Fresh frozen PDAC samples and whole blood samples	WGS, RNA-seq, RNA-ISH	[72]
Maurer et al., 2019	Molecular subtypes of PDAC	Fresh frozen PDAC samples	RNA sequencing	[71]
Nadella et al., 2019	Role of gastrin in stimulating KRAS and in turn carcinogenesis	Gastrin Knockout mice	Reverse phase protein array, IHC, miRNA analysis	[66]
Hiroshima et al., 2019	Impact of FN1-ITGA3 on prognosis of PDAC	Fresh frozen tissue of PDAC	LC-MS/MS	[93]
Robin et al., 2020	Prognostic role of stratifin	PDAC FFPE samples	Gene expression analysis, IHC, ELISA	[83]
Birnbaum et al., 2021	Transcriptomic analysis of PDAC samples to identify molecular subtypes of PDAC	Fresh frozen PDAC samples	RNA-seq, RNA-ISH	[34]
Kalloger et al., 2021	Prognostic roles of genes expressed in stroma and epithelium of PDAC	PDAC FFPE samples	mRNA quantification	[70]

The hypoxic environment of PC is another widely investigated area. During cancer progression, the cells undergo rapid proliferation resulting in consuming a huge amount of oxygen. The drastic alteration in the oxygen levels stimulates a number of proteins such as Insig2, HIF1A, and BNIP3, which in turn activates the downstream pathways, which leads to more aggressive behavior and therapy resistance in PC (Figure 3) [94,96,97,98,99]. Hypoxia-inducible factors (HIFs) are heterodimeric transcription factors made of two subunits, alpha and beta (HIFα and HIFβ) [98]. HIFα is known to induce the *VEGF* (vascular endothelial growth factor), *PDGFA* (platelet-derived growth factor alpha), and *FGF2* (coding basic fibroblast growth factor, bFGF), but has not been explored much [96,99]. *HIF1A* induces the glycolytic enzymes as well, the *PGK1* (phosphoglycerate kinase 1) is one such enzyme found overexpressed in microdissected PDAC samples analyzed using proteomic studies. They also marked its potential to act as a diagnostic biomarker or as a therapeutic target [95]. A study conducted on microdissected FFPE samples of PDAC by qPCR and other statistical analysis showed the correlation between the genes *HIF1A*, *FGF2*, *VEGF*, and *PDGFA* in PC development and the significance of *HIF1A* in prognosis [96]. Inspired by recent research on insulin-induced gene 2 (*INSIG2*) as a novel biomarker for colon cancer, Kayashima et al. attempted to study the involvement of *INSIG2* in pancreatic malignancy. They analyzed *INSIG2* mRNA expression on laser microdissected normal pancreatic epithelial cells, invasive ductal carcinoma cells, and PanIN cells, as well as on PC cell lines cultured under normoxic (21% O_2_) and hypoxic (<1% O_2_) conditions. They found a significant increase of *INSIG2* expression in the PC cell line under the hypoxic conditions as well as in the microdissected samples. Cell proliferation and invasion were found to be decreased in one of the PC *INSIG2*-knockdown cell lines. The mRNA expression levels were also evidently higher in late-stage cancer compared to the early stage [97]. The hypoxia-inducible proapoptotic gene, *BNIP3,* was discovered to be downregulated in PDAC tissues as well as in cell lines. It showed resistance to both drugs gemcitabine and 5-fluorouracil, which led to a lower patient survival rate and a worse prognosis [94]. All these appealing results pointed out that these proteins play a vital role in pancreatic cancer progression and metastasis and they can act as a biomarker for diagnosis, prognosis, and therapy.

The ultimate goal of all cancer study efforts is to find a suitable solution for the disease’s management or cure. Having said about the aggressive nature, poor prognosis, and decreased survival rate of PC, the studies on prognosis, diagnosis, therapy, and survival are of great importance both to physicians as well as to the public. Several studies are being conducted using bulk tissues as well as on specific cell populations to understand the scenario better. The importance of LCM lies as it aids in isolating the specific cell populations of interest and allows the research work carried out in a specific direction.

## 4. Conclusions

Pancreatic cancer is the rarest and most aggressive disease with a poor prognosis. The development of techniques for diagnosis, prognosis, and therapy is of the most importance. LCM is a powerful tool that has the capability of isolating specific cell populations from FFPE, fresh frozen, and cell-cultured samples. Utilizing LCM facilitated the isolation of a particular cell population and the discovery of some excellent findings, such as the involvement of gene/protein alterations in the downstream pathway leading to PC developments. These findings are critical for comprehending prognostic/diagnostic biomarkers and addressing potential therapeutic targets. PC’s stromal and cancer subtypes were better categorized using the LCM, which provided researchers with a holistic image of the tumor microenvironment. Various high throughput methods were employed downstream of LCM to interpret different areas of cancer. Improving the staining methods, laser systems, and preservation methods would improve LCM’s scope in cancer research. More could be unravelled using this technique or any other single-cell separating techniques such as FACS and microfluidics platforms, as it allows us to study the individual cells or groups of cells with the same characteristics, giving us more accurate results. This could aid in discovering a potential biomarker for prognosis, diagnosis, or even identifying a therapeutic target. In the future, we must contribute more towards implementing them into clinical practice and, consecutively, to society.

## Figures and Tables

**Figure 1 ijms-23-14566-f001:**
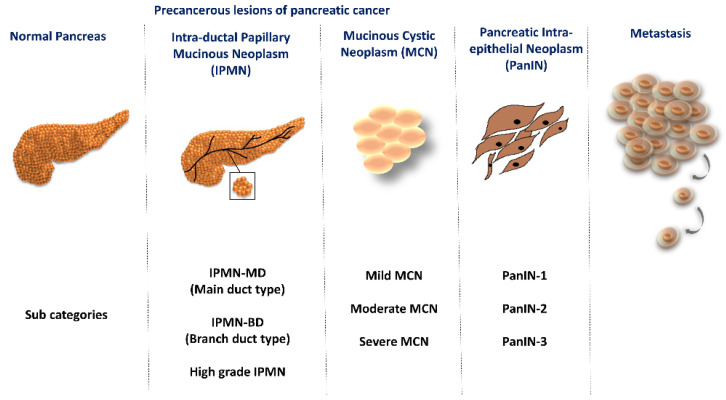
Site of precancerous lesions of pancreatic cancer and their sub-categories.

**Figure 3 ijms-23-14566-f003:**
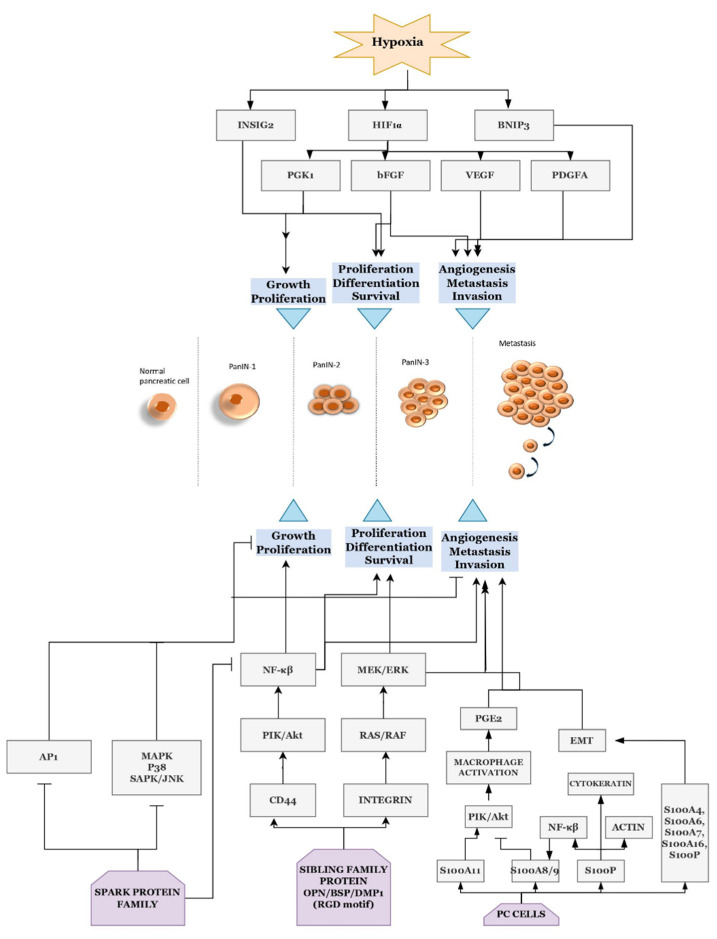
Schematic diagram of the possible mechanism of tumorigenesis of PC by SIBLING protein family, SPARC protein family, and S100 protein family, and proteins stimulated during hypoxia condition resulting in cancer development. SIBLING, Small Integrin Binding Ligand N-linked Glycoprotein; OPN, Osteopontin; BSP, Bone SialoProtein; DMP1, Dentin Matrix Protein I; CD44, Cluster of Differentiation 44; RAS/RAF, RAt Sarcoma/Rapidly Accelerated Fibrosarcoma GTP binding proteins; MEK/ERK, Mitogen-Activated protein kinases/Extracellular signal-regulated protein kinases; PI3K/Akt, Phosphatidylinositol 3-Kinase/Ser/Thr Protein kinase; NF-κβ, Nuclear factor Kappa light-chain-enhancer of activated B cells; MAPK, Mitogen-Activated Protein Kinase; SAPK/JNK, Stress-activated protein kinases/c-Jun N-terminal Kinases; AP-1, Activating protein-1; PC cells, Pancreatic Cancer cells; PGE2, Prostaglandin E2; EMT, Epithelial to Mesenchymal Transition; INSIG2, Insulin-induced gene 2; HIF1α, Hypoxia-induced factor-1α; BNIP3, BCL2 interacting protein 3; PGK1, phosphoglycerate kinase 1; bFGF, basic fibroblast growth factor; VEGF, vascular endothelial growth factor; PDGFA, Platelet-derived growth factor-alpha.

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
