# Peer review of "Laser Capture Microdissection: A Gear for Pancreatic Cancer Research"

_ijms, 2022, doi:10.3390/ijms232314566_

Round 1
Reviewer 1 Report
B.H Rao reported a review on laser capture microdissection which is a strong tool to help pancreatic cancer. Mechanism and impact were concluded with details. This is an interesting and well-structured work and provides good instruction for LCM. Based on that, acceptance in the present form is recommended.
Author Response
We thank Reviewer 1 for a very positive comment. It is really encouraging that R1 believes that our review provided good instructions of LCM and that we concluded the impact of LCM in pancreatic cancer research.
Reviewer 2 Report
The manuscript entitled “Laser capture microdissection: A gear for pancreatic cancer research” by Bhavana Hemantha Rao et al. presented a review on the laser capture microdissection (LCM) technique used in pancreatic cancer research.
This review manuscript is well-organized and easy to follow. The authors presented basic technique background about LCM, then reviewed the applications of LCM on pancreatic cancer research.
To address the following comments, I recommend a minor revision of the manuscript before publication:
(1) To improve the readability, please consider separating more paragraphs for sections “3.1. Mutation studies”, “3.2. Breakthrough of PC subtypes and their relevance in survival”, and “3.3. Proteins, pathways, and cancer management”.
(2) There is a typo on line 270, please add a space between “[64]” and “found”.
(3) Please improve the resolution of Figure 3.
(4) Please consider adding explanations for Table 1 in the main text. Currently, Table 1 is only mentioned on line 385.
(5) I suggest the authors to add more discussions about advantages and disadvantages of LCM, comparing with FACS and microfluidics technologies.
Author Response
We would like to thank Reviewer 2 for encouraging comments and valuable suggestions. We have addressed all suggested comments and improved the quality of the manuscript. Responses to individual comments are provided below:
1/ We thank the reviewer for this important suggestion. We agree that the readability will be improved by breaking the sections into more paragraphs. We separated the sections 3.1, 3.2 and 3.3 into 11 paragraphs.
2/ We corrected the typo on Line 299 in the revised manuscript.
3/ High-resolution Figure 3 was included in the manuscript.
4/ We agree with the reviewer. Table 1 was referenced on Line 217 and further discussed and explained in detail on Lines 294-297 and 419-426.
5/ We would like to thank the reviewer for this suggestion. We agree that the manuscript would benefit from more discussion on comparison of LCM with alternative techniques. We thoroughly discussed advantages and disadvantages of LCM and compared them with droplet-based microfluidics and FACS. In addition, we included five new references in the revised manuscript. Additional discussion was added on lines 96, 114-119, 157-158 and 162-170. New references [27-29] and [38-39] were added.